# Systematic review and meta-analysis of cohort studies of long term outdoor nitrogen dioxide exposure and mortality

David M. Stieb[1,2☯]*, Rania Berjawi[2☯], Monica Emode[3☯], Carine Zheng[2¤a‡], Dina Salama[2¤b‡], Robyn Hocking[4‡], Ninon Lyrette[5¤c‡], Carlyn Matz[5‡], Eric Lavigne[2,5‡], Hwashin H. Shin[1,6‡]

1 Environmental Health Science and Research Bureau, Health Canada, Ottawa, Ontario, Canada, 2 School of Epidemiology and Public Health, University of Ottawa, Ottawa, Ontario, Canada, 3 School of Population and Public Health, University of British Columbia, Vancouver, British Columbia, Canada, 4 Learning, Knowledge and Library Services, Health Canada, Ottawa, Ontario, Canada, 5 Water and Air Quality Bureau, Health Canada, Ottawa, Ontario, Canada, 6 Department of Mathematics and Statistics, Queen's University, Kingston, Ontario, Canada

☯ These authors contributed equally to this work.
¤a Current address: Center for Income and Socio-Economic Wellbeing, Statistics Canada, Ottawa, Ontario, Canada
¤b Current address: Department of Family Medicine, University of British Columbia, Vancouver, Canada
¤c Current address: Chemicals and Environmental Health Management Bureau, Health Canada, Ottawa, Ontario, Canada
‡ CZ, DS, RH, NL, CM, EL and HHS also contributed equally to this work.
* dave.stieb@canada.ca

**Data Availability Statement:** All relevant data are within the manuscript and its Supporting Information files.

## Abstract

### Objective

To determine whether long term exposure to outdoor nitrogen dioxide ($NO_2$) is associated with all-cause or cause-specific mortality.

### Methods

MEDLINE, Embase, CENTRAL, Global Health and Toxline databases were searched using terms developed by a librarian. Screening, data extraction and risk of bias assessment were completed independently by two reviewers. Conflicts were resolved through consensus and/or involvement of a third reviewer. Pooling of results across studies was conducted using random effects models, heterogeneity among included studies was assessed using Cochran's Q and $I^2$ measures, and sources of heterogeneity were evaluated using meta-regression. Sensitivity of pooled estimates to individual studies was examined and publication bias was evaluated using Funnel plots, Begg's and Egger's tests, and trim and fill.

### Results

Seventy-nine studies based on 47 cohorts, plus one set of pooled analyses of multiple European cohorts, met inclusion criteria. There was a consistently high degree of heterogeneity. After excluding studies with probably high or high risk of bias in the confounding domain (n = 12), pooled hazard ratios (HR) indicated that long term exposure to $NO_2$ was significantly

**Funding:** The study was funded by Health Canada under operating funding. There is no grant number associated with the funding. Authors DMS, RH, NL, CM, EL and HHS receive a salary from the funder. Authors RB, ME, CZ and DS were employed as students by the funder. The funders had no role in study design, data collection and analysis, decision to publish, or preparation of the manuscript.

**Competing interests:** The authors have declared that no competing interests exist.

associated with mortality from all/ natural causes (pooled HR 1.047, 95% confidence interval (CI), 1.023–1.072 per 10 ppb), cardiovascular disease (pooled HR 1.058, 95%CI 1.026–1.091), lung cancer (pooled HR 1.083, 95%CI 1.041–1.126), respiratory disease (pooled HR 1.062, 95%CI1.035–1.089), and ischemic heart disease (pooled HR 1.111, 95%CI 1.079–1.144). Pooled estimates based on multi-pollutant models were consistently smaller than those from single pollutant models and mostly non-significant.

## Conclusions

For all causes of death other than cerebrovascular disease, the overall quality of the evidence is moderate, and the strength of evidence is limited, while for cerebrovascular disease, overall quality is low and strength of evidence is inadequate. Important uncertainties remain, including potential confounding by co-pollutants or other concomitant exposures, and limited supporting mechanistic evidence. (PROSPERO registration number CRD42018084497)

## Introduction

Traditional cohort studies involving recruitment of individual participants and long term follow-up over many years have been foundational in linking long term air pollution exposure to mortality [1, 2]. However, these studies are time-consuming and expensive, and may not be representative of the general population depending on recruitment procedures and response rates. More recently, large "synthetic" cohorts have been established by linking administrative or health survey data with address and mortality data [3–5]. Compared to traditional cohort studies, these newer cohort studies have the advantages of being less expensive and time-consuming, more statistically powerful and more representative of the general population. However, they potentially lack data on important covariates such as smoking, which could confound the association between air pollution and mortality. Both types of cohort studies have benefited from innovations in exposure assessment, including utilization of remote sensing, chemical/meteorological and dispersion models, as well as land use regression models [6–8]. These innovations have revolutionized exposure assessment, reducing reliance on sparse networks of ground monitors and increasing both geographic coverage and spatial resolution of estimated exposures.

With increasing exposure assessment capabilities, interest has grown in specific air pollution sources, including traffic. Nitrogen dioxide ($NO_2$) is a commonly employed marker of traffic-related urban air pollution [9, 10], although it also more broadly reflects any combustion in air, from sources such as industry and fossil fuel powered electric power generating stations [11, 12]. Ambient concentrations of $NO_2$ have declined considerably over the last 15–20 years in North America, Europe, Japan and South Korea, but concentrations are increasing in other areas (e.g. China, North Korea and Taiwan) [13]. Numerous studies have evaluated health effects of $NO_2$ on diverse body systems. However, whether long term $NO_2$ exposure is causally related to mortality remains an unresolved question. A particular complicating factor is whether $NO_2$ itself is to blame, or whether it is simply acting as a marker for specific air pollution sources i.e. emissions from vehicles, or more generally as a marker for spatial variation in the urban air pollution mixture [9]. Carbon monoxide and certain chemical components of fine particulate matter, also primarily originating from vehicle emissions, are key potential confounders, given their well-established pathophysiological mechanisms of action [14].

Effects of $NO_2$ could also be confounded by other concomitant traffic-related exposures such as noise or stress [15].

The aforementioned combination of methodological developments has led to rapid growth in the number and geographic diversity of cohort studies, and makes it timely to synthesize the available evidence. Five previous systematic reviews/ meta-analyses have evaluated the association of long term $NO_2$ exposure and mortality [16–20]. However, only one of these reviews conducted an assessment of risk of bias over multiple domains, none provided pooled estimates from paired single and multi-pollutant models from the same primary studies, and only studies published up to 2018 were included. Our objective is therefore to determine whether long term exposure to outdoor $NO_2$ is associated with all-cause or cause-specific mortality based on an up to date synthesis of the available evidence.

## Methods

The protocol is registered with PROSPERO (CRD42018084497) (S1 File) [21].

### Literature searches

MEDLINE, Embase, CENTRAL, Global Health and Toxline databases were searched using terms developed by a librarian (S1 Table). The search strategy underwent Peer Review of Electronic Search Strategies (PRESS) [22]. Searches were last updated February 25, 2020. Inclusion criteria were as follows: Participants/population: Humans; Intervention(s), exposure(s): Exposure to outdoor $NO_2$ (and other oxides of nitrogen); Comparator(s)/control: Lower levels of exposure; Main outcomes: Mortality from all/ natural causes or specific causes; Study design: cohort. Publications in abstract form only were excluded. Publications in English or French were included and there were no restrictions on publication date. Effect measures considered were: mortality effects reported as regression coefficients, hazard ratios (HR) or relative risks associated with exposures over a period of years, expressed per specified increment in exposure. The present review is one part of a series of reviews of effects of $NO_2$, all of which were included in the original search. Other reviews pertain to non-asthma respiratory morbidity and ischemic heart disease morbidity related to short term exposure [21]. Studies were selected for the present review if reported outcomes matched the inclusion criteria specified above.

### Screening, data extraction and risk of bias assessment

Screening and data extraction were completed independently by two reviewers in DistillerSR. Conflicts between reviewers were resolved through consensus and/or involvement of a third reviewer. All studies retrieved from literature searches were screened for relevance based on title and abstract according to the above inclusion criteria. Where relevance could not be determined based on abstract and title, the full text was reviewed. Manual searches were also completed of reference lists of all relevant studies. Bibliographic data, study location and timing, population age group(s), sample size, cause of death (including the International Classification of Diseases (ICD) code(s) if available), method of exposure assessment, pollutant (including name, units, descriptive statistics), type of regression model, effect measure and standard error or confidence interval, model covariates (potential confounders) and their specification were extracted from all studies meeting inclusion criteria. When single pollutant results were presented for multiple exposure periods, we extracted the most highly statistically significant result (regardless of the direction of the association), or that reported by the authors as their primary finding. Results from multi-pollutant models that resulted in the greatest reduction in magnitude of effect compared to single pollutant results were selected in order to bracket the magnitude of effect from each study. Our objective in this regard was not to assess

the magnitude of the association with $NO_2$ in the presence of a single co-pollutant common to all studies, but rather to determine the maximum potential for confounding of the association of mortality with $NO_2$ by any co-pollutant(s). Results expressed per pollutant increment expressed in μg/m$^3$ were converted to parts per billion [23], and those based on nitrogen oxides ($NO_x$) were converted by multiplying the log(HR) by 2.31 (the average ratio of log(HR) based on $NO_2$ to log(HR) based on $NO_x$ in four studies based on 25 cohorts [24–27]). Where required data were not provided, authors were contacted by e-mail. In some instances Engauge Digitizer [28] was employed to extract numeric results presented only in graph form. The Navigation Guide systematic review methodology [29] was employed to evaluate risk of bias at the study level according to the following domains: selection bias, exposure assessment, confounding, outcome assessment, completeness of outcome data, selective outcome reporting, conflict of interest (including sources of funding) and other sources of bias. Criteria are detailed in S1 Fig. In the confounding domain, we considered age, sex, smoking and socioeconomic status as critical potential confounders. As a sensitivity analysis, we treated body mass index (BMI) rather than smoking as a critical confounder, as specified in the World Health Organization (WHO) risk of bias criteria [30]. Assessment of risk of bias was completed independently by two reviewers and conflicts between reviewers were resolved through consensus and/or involvement of a third reviewer.

## Data analysis

Pooling of HRs across studies was conducted using random effects models computed using Restricted Maximum Likelihood (REML) estimation, with sensitivity analyses employing Dersimonian and Laird and Empirical Bayes estimators [31]. Heterogeneity among included studies was assessed using Cochran's Q and $I^2$ measures, and sources of heterogeneity were evaluated using meta-regression [31]. Sensitivity of pooled estimates to individual studies was examined using Leave One Out analysis and publication bias was evaluated using Funnel plots, Begg's and Egger's tests, and trim and fill [31]. Subgroup analyses were conducted by cause of death, region, risk of bias characterization (pre-specified in protocol) and single vs. multi-pollutant models. Analysis was conducted in R version 3.6.0 [32] using the metafor package [31].

## Overall rating of quality and strength of evidence

We applied the Navigation Guide methodology [33] and the causality determination framework used by the US EPA and Health Canada [12] (S2 and S3 Tables) to judge the overall quality and strength of the evidence, and likelihood of a causal relationship. Following the Navigation Guide methodology, given the observational nature of the evidence, the starting point for rating overall quality was "moderate," which was downgraded or upgraded to "low" or "high" based on the criteria summarized in S2 Table [33]. Identification of multiple upgrading or downgrading factors does not necessarily mean that overall quality is upgraded or downgraded multiple levels. Rather, the overall degree of upgrading or downgrading is based on reviewer judgement of the overall importance and impact of all factors considered [33]. The Navigation Guide characterizes strength of evidence of toxicity as "sufficient", "limited", "inadequate" or "indicative of a lack of toxicity" (S2 Table), based on the overall quality of the evidence, the direction of effect, confidence in the effect and any other factors identified as germane by the reviewers [33]. Given the parallels with the USEPA/Health Canada causality determination criteria (S3 Table), while we did not conduct a systematic review of other lines of evidence, we drew upon summaries of other lines of evidence from a recent assessment

document [11], supplemented by findings from more recent mechanistic studies, in order to characterize the likelihood of a causal relationship.

## Results

A Preferred Reporting Items for Systematic Reviews and Meta-Analyses (PRISMA) diagram summarizing disposition of studies identified in literature searches is shown in Fig 1. As indicated earlier, the present review is one part of a series of reviews of effects of $NO_2$ on multiple outcomes, all of which were included in the original search, which is reflected in numeric results reported in Fig 1. Seventy-nine studies were included in our final analysis based on 47 cohorts, plus one set of pooled analyses of multiple European cohorts. Study characteristics are summarized in S4 Table. Approximately the same proportion of studies was conducted in North America (n = 32, 41%) and Europe (n = 35, 44%), while a smaller proportion (n = 12, 15%) was conducted elsewhere. Cohorts were mostly drawn from the general population, with the exception of six cohorts based on mortality follow-up after diagnosis of hypertension [34–38], myocardial infarction [39–41], stroke [42, 43], or lung cancer [26, 44]. Cohort sizes ranged

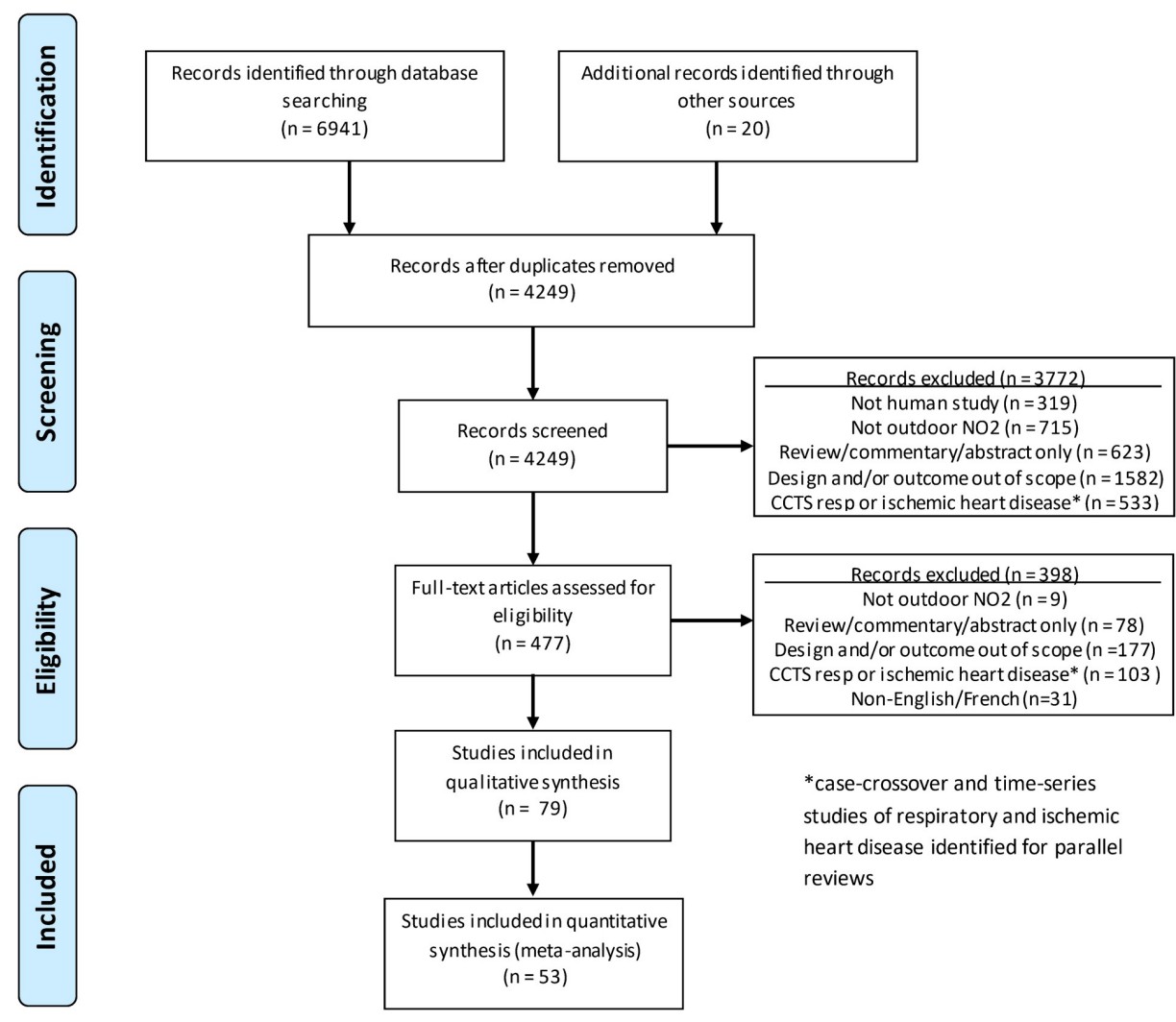

**Fig 1. Preferred Reporting Items for Systematic Reviews and Meta-Analyses (PRISMA) flow diagram.**

from 1,800 to 14.1 million participants. Fifty-one studies (65%) employed modelling while 28 (35%) employed ground monitoring as the source of exposure data, and almost all, n = 74 (94%), employed $NO_2$ as the exposure metric, while only 5 (6%) employed $NO_x$. The most frequently examined causes of death were all/natural cause (non-accidental), (n = 56, 71%), cardiovascular (n = 37, 47%), respiratory (n = 35, 44%), lung cancer (n = 32, 41%), ischemic heart disease (n = 27, 34%), and cerebrovascular (n = 26, 33%). Thirty-eight studies (48%) were mostly conducted post 2000 (majority of study duration after 2000).

Risk of bias ratings for individual studies are shown in S2 Fig and reasons for assigned ratings of risk of bias greater than low risk (or unable to assess) for individual studies are provided in S5 Table. The greatest variability in ratings occurred in the exposure assessment and confounding domains, while ratings in the other six domains (selection bias, outcome assessment, completeness of outcome data, selective outcome reporting, conflict of interest, other sources of bias) were generally low or probably low risk of bias. Forty-six studies (58.2%) were rated probably high or high risk of bias or unable to assess in the exposure assessment domain (although bias would be expected to be non-differential) because exposure was based on an area-level average, did not account for change of address, relied on a single monitor, there was evidence of a mediocre correlation of modelled or measured values with ground measurements in the target community, or there was insufficient information. Twelve studies (15.2%) were rated probably high or high risk of bias in the confounding domain because of lack of adjustment for one or more key potential confounders (age, sex, smoking, socio-economic status). An additional eight studies did not adjust for BMI or account for obesity as a potential confounder through other covariates such as comorbidity or analysis of associations with exposure in complementary data.

## Effect estimates and pooled effect estimates

All effect estimates from individual studies, including from single and multi-pollutant models, and by population and outcome subgroup are provided in forest plots by region in S3 Fig. Of these, we excluded estimates from pooling if they were superseded by other studies encompassing the same geographic area or time period, e.g. in subsequent multi-city studies or those spanning a longer study duration, results were provided only from multi-pollutant models, or there were too few studies of the outcome, leaving 53 studies included in the meta-analysis (see S4 Table). Pooled estimates based on single pollutant model results by cause of death are summarized in Table 1. Pooled estimates were largest for respiratory and cerebrovascular causes of death, similar for cardiovascular and ischemic heart disease, and smallest for all/natural cause and lung cancer. All pooled estimates were characterized by a high degree of heterogeneity.

## Meta-regression and sensitivity analysis

Meta-regression revealed that the magnitude of the log HR for cardiovascular, respiratory and ischemic heart disease mortality was significantly larger for studies conducted outside North America and Europe (p = 0.002, 0.025, 0.021 respectively). For lung cancer mortality, standard deviation (p = 0.0009) and range (p = 0.006) of $NO_2$ exposure were significantly positively associated with log HR. Probably high or high risk of bias in the confounding domain was significantly positively associated with log HR for all/ natural cause (p = 0.040), respiratory (p = 0.006) and cerebrovascular (p = 0.036) mortality. In a sensitivity analysis, treating BMI rather than smoking as a critical potential confounder, as specified in the WHO risk of bias criteria, probably high or high risk of bias in the confounding domain was not associated with log HR. Risk of bias in the exposure assessment domain, study interquartile range $NO_2$ and timing of study primarily before or after 2000 were not significant predictors of the log HR. Residual

**Table 1. Pooled estimates by cause of death and sensitivity analyses.**

| Cause of death | N | P(Q) | I² (%) | Pooled HR per 10 ppb | Lower 95% CI | Upper 95% CI |
|---|---|---|---|---|---|---|
| **All/ Natural causes** | 39 | <0.0001 | 99.3 | 1.064 | 1.034 | 1.094 |
| **Cardiovascular** | 29 | <0.0001 | 99.9 | 1.139 | 0.997 | 1.301 |
| **Respiratory** | 29 | <0.0001 | 99.7 | 1.174 | 1.016 | 1.357 |
| **Lung cancer** | 28 | <0.0001 | 96.6 | 1.084 | 1.045 | 1.124 |
| **Ischemic heart disease** | 19 | <0.0001 | 96.1 | 1.128 | 1.076 | 1.182 |
| **Cerebrovascular** | 17 | <0.0001 | 99.7 | 1.167 | 0.936 | 1.456 |
| Excluding studies with probably high or high risk of bias in confounding domain | | | | | | |
| **All/ Natural causes** | 32 | <0.0001 | 96.7 | 1.047 | 1.023 | 1.072 |
| **Cardiovascular** | 23 | <0.0001 | 92.8 | 1.058 | 1.026 | 1.091 |
| **Respiratory** | 24 | <0.0001 | 65.9 | 1.062 | 1.035 | 1.089 |
| **Lung cancer** | 23 | <0.0001 | 85.1 | 1.083 | 1.041 | 1.126 |
| **Ischemic heart disease** | 14 | 0.0001 | 69.9 | 1.111 | 1.079 | 1.144 |
| **Cerebrovascular** | 13 | 0.0738 | 0.1 | 1.014 | 0.997 | 1.032 |
| Excluding studies with probably high or high risk of bias in confounding domain and exposure based on $NO_x$ | | | | | | |
| **All/ Natural causes** | 29 | <0.0001 | 92.4 | 1.033 | 1.016 | 1.051 |
| **Cardiovascular** | 21 | <0.0001 | 94.3 | 1.056 | 1.020 | 1.093 |
| **Respiratory** | 21 | <0.0001 | 67.6 | 1.057 | 1.031 | 1.085 |
| **Lung cancer** | 21 | <0.0001 | 85.6 | 1.075 | 1.033 | 1.119 |
| **Ischemic heart disease** | 12 | 0.0003 | 56.7 | 1.104 | 1.078 | 1.131 |
| **Cerebrovascular** | 11 | 0.0488 | 0.5 | 1.013 | 0.996 | 1.032 |
| Excluding studies with probably high or high risk of bias in confounding domain (WHO criteria–BMI rather than smoking as critical potential confounder) | | | | | | |
| **All/ Natural causes** | 31 | <0.0001 | 98.4 | 1.063 | 1.027 | 1.100 |
| **Cardiovascular** | 19 | <0.0001 | 98.1 | 1.097 | 1.025 | 1.173 |
| **Respiratory** | 18 | 0.0129 | 0.1 | 1.041 | 1.028 | 1.054 |
| **Lung cancer** | 17 | <0.0001 | 89.7 | 1.082 | 1.0279 | 1.134 |
| **Ischemic heart disease** | 14 | <0.0001 | 91.4 | 1.149 | 1.093 | 1.208 |
| **Cerebrovascular** | 13 | <0.0001 | 96.3 | 1.080 | 0.973 | 1.200 |

heterogeneity remained high (I² >80%) even after accounting for significant predictor variables.

Based on meta-regression findings, studies with probably high or high risk of bias in the confounding domain (n = 12) were excluded, resulting in substantially smaller pooled HRs for cardiovascular, respiratory and cerebrovascular mortality (Table 1). Pooled estimates were significantly smaller for cerebrovascular mortality (p = 0.031) and significantly larger for ischemic heart disease mortality (p = 0.002), than for all/ natural cause mortality. Fig 2 presents a forest plot of HRs from individual studies and pooled estimates for all/ natural cause mortality, after exclusion of studies with probably high or high risk of bias in the confounding domain. The pooled estimate for North America was smaller than those for Europe and other countries, and heterogeneity among primary studies was lower for North American studies. Forest plots for other causes of death are provided in S4 Fig. Applying the WHO risk of bias criteria (treating BMI rather than smoking as a critical potential confounder) resulted in a somewhat larger pooled estimate for all/natural cause mortality, considerably larger pooled estimates for cardiovascular and cerebrovascular mortality, together with much greater heterogeneity for the latter, and a somewhat smaller pooled estimate with narrower confidence intervals and considerably less heterogeneity for respiratory mortality (Table 1).

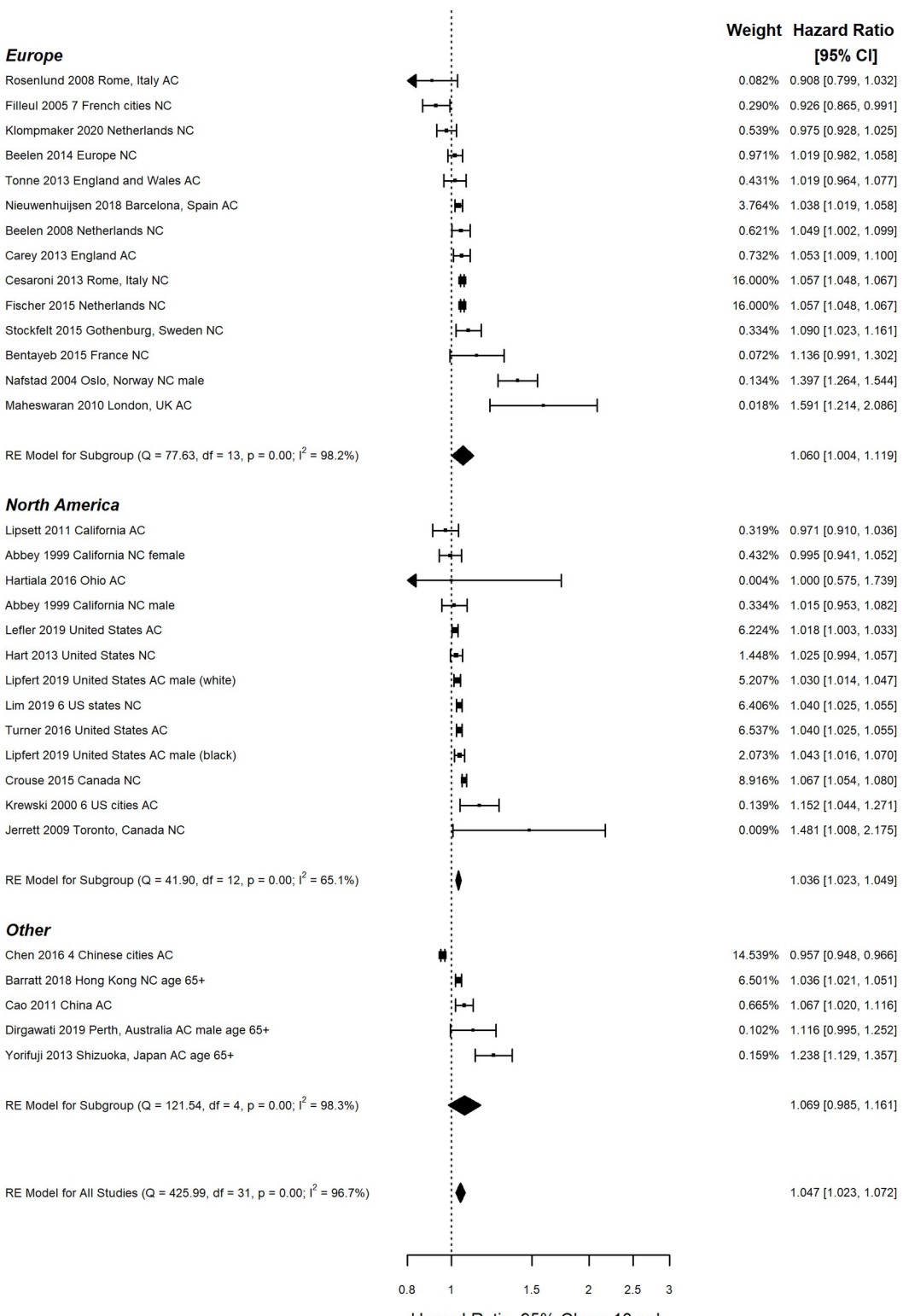

**Fig 2. Hazard ratios from single pollutant models from individual cohort studies and pooled estimates by region, for all/ natural cause mortality (AC/NC).**

Results were generally insensitive to the exclusion of studies employing $NO_x$ as the exposure metric [45–47], with the exception of all/natural cause mortality, for which the pooled estimate was somewhat smaller (Table 1). Findings were also insensitive to excluding cohorts of individuals with pre-existing disease [36, 39, 41, 42] (pooled estimate of HR for all/natural cause mortality 1.050, 95% CI 1.022–1.078). Pooled estimates were not sensitive to employing alternative estimators except that confidence intervals tended to be slightly narrower based on Dersimonian and Laird, and slightly wider based on Empirical Bayes (S1 Text). In leave one out analyses, heterogeneity among respiratory and ischemic heart disease studies was sensitive to exclusion of results from Katanoda et al. [48] and Turner et al. [49] respectively (S6 Table). Egger's test revealed significant funnel plot asymmetry for all/ natural cause (p = 0.0022) and lung cancer (p = 0.031) mortality, while Begg's test indicated significant funnel plot asymmetry only for lung cancer mortality (p = 0.022). Trim and fill analysis estimated that there were 4 missing studies with log HR below the pooled estimate for lung cancer mortality (S5 Fig), reducing the pooled estimate slightly to 1.066 (95%CI 1.024–1.110).

## Effects of co-pollutants, noise and green space

A forest plot of nine paired estimates of associations with all/ natural cause mortality from single and multi-pollutant models from the same study, after exclusion of studies with probably high or high risk of bias in the confounding domain, is shown in Fig 3. The pooled estimate from single pollutant models was higher than that from multi-pollutant models with 1–4 co-

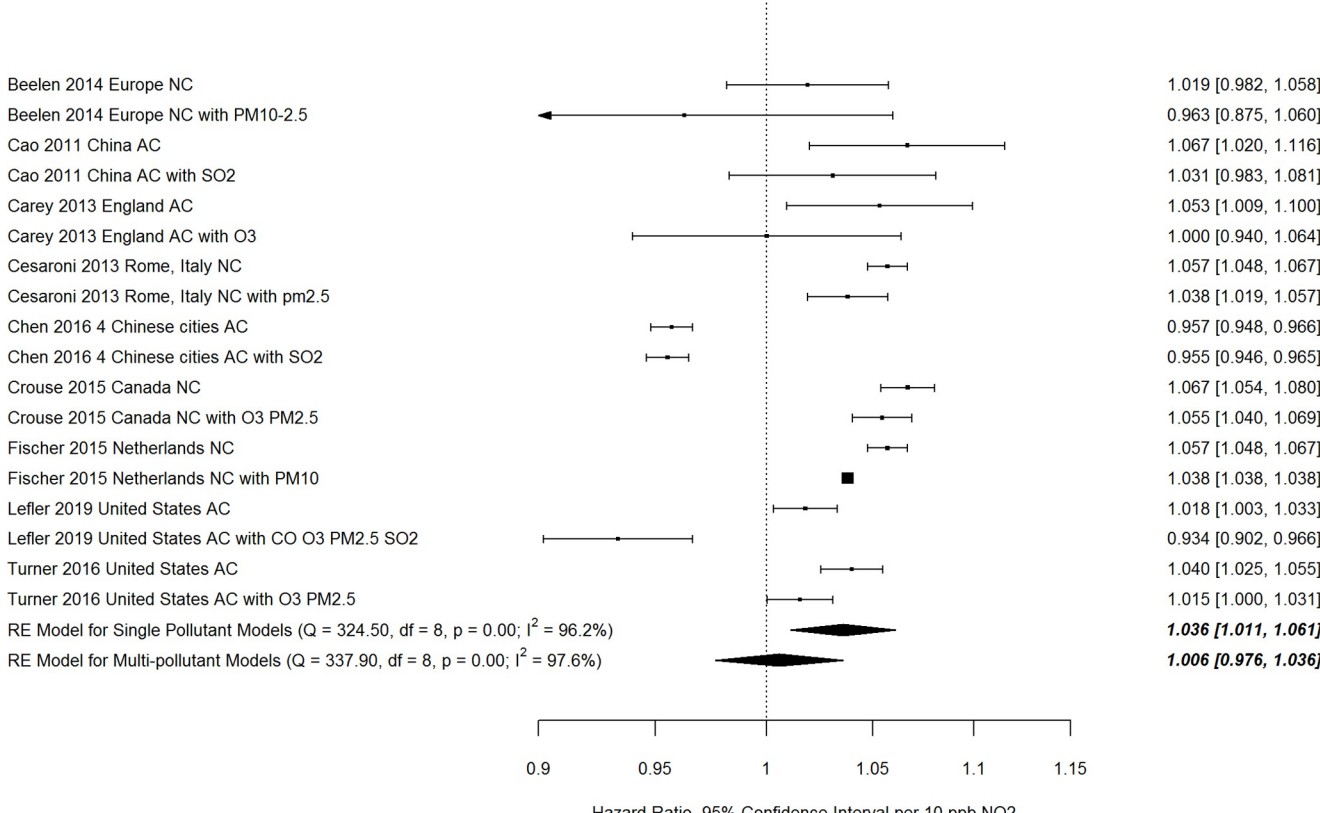

**Fig 3. Hazard ratios from individual cohort studies and pooled estimates from single and multi-pollutant models for all/ natural cause mortality (AC/NC).**

pollutants, and the confidence interval for the multi-pollutant pooled estimate included 1, indicating no association. However, the difference between pooled estimates for single and multi-pollutant models was not significant (p = 0.13). Fewer paired estimates from single and multi-pollutant models were available for other causes of death. Pooled estimates from multi-pollutant models were consistently smaller in magnitude than those from single pollutant models, and the pooled estimates based on multi-pollutant models included 1, indicating no association, with the exception of ischemic heart disease mortality which was based on only three studies (Table 2).

Few studies have jointly examined effects of $NO_2$ and noise. Sørensen et al. found that $NO_2$ exhibited a significant positive association with fatal stroke when modelled jointly with traffic noise [50], while in the same cohort, Hvidtfeldt et al. found that $NO_2$ remained significantly associated with natural cause mortality after adjustment for traffic noise, although the magnitude of the association was reduced, and $NO_2$ was no longer significantly associated with cardiovascular mortality after adjustment for traffic noise [51]. Héritier et al. [52] also reported that the association of $NO_2$ with myocardial infarction mortality was no longer significant after adjustment for traffic, rail and aircraft noise. Klompmaker et al. found no association between $NO_2$, $PM_{2.5}$, greenness or traffic noise and mortality [53], and Tonne et al. observed a negligible change in the association between $NO_2$ and mortality with additional adjustment for traffic noise [40]. Nieuwenhuijsen et al. [54] found that the association of $NO_2$ with mortality was reduced in magnitude in models with noise and/or green space.

## Shape of exposure-response relationship

Twenty-nine studies evaluated the shape of the exposure-response relationship between $NO_2$ or $NO_x$ and mortality by examining the association by pollutant quantile [24, 41, 44, 47, 55–57], plotting the association using a non-linear function of pollutant concentration and/or testing the significance of the difference between linear and non-linear models [5, 25, 27, 45, 46, 50, 51, 53, 57–69], or plotting associations by city [48]. Of these, 19 studies found a linear association [27, 41, 44, 50, 51, 53, 55–57, 60, 63–65, 67–69], in some instances only in subsets of the data by cause of death [5, 45, 46], and five studies concluded that the association was supra-linear [25, 45, 47, 59, 66]. Three found evidence of a threshold [48, 61, 62], but only one study identified specific numeric values, ranging from 20–40 $\mu g/m^3$, depending on age and cause of death [62]. Two studies reported no association, based on analysis by quantiles [24], and using a non-linear function of $NO_2$ [58]. While there was some variation in findings by cause of

**Table 2. Pooled estimates from single and multi-pollutant models by cause of death.**

| Cause of death | Model | n | Pooled HR (95% CI) | p (difference) |
|---|---|---|---|---|
| All/natural cause | Single | 9 | 1.036 (1.011–1.061) | 0.13 |
| | Multi | 9 | 1.006 (0.976–1.036) | |
| Cardiovascular | Single | 4 | 1.053 (1.011–1.096) | 0.26 |
| | Multi | 4 | 1.018 (0.975–1.063) | |
| Lung cancer | Single | 5 | 1.057 (0.967–1.156) | 0.65 |
| | Multi | 5 | 1.028 (0.948–1.115) | |
| Respiratory | Single | 5 | 1.039 (1.024–1.054) | 0.21 |
| | Multi | 5 | 1.013 (0.977–1.051) | |
| Cerebrovascular | Single | 2 | 1.000 (0.976–1.025) | 0.67 |
| | Multi | 2 | 0.977 (0.879–1.086) | |
| Ischemic heart disease | Single | 3 | 1.106 (1.064–1.150) | 0.21 |
| | Multi | 3 | 1.071 (1.038–1.105) | |

death in individual studies, overall there was evidence of an exposure-response relationship across all six causes of death.

## Discussion

Based on an analysis of 53 cohort studies, pooled estimates of associations indicated that long term exposure to outdoor $NO_2$ was significantly associated with mortality from all/ natural causes, cardiovascular disease, lung cancer, respiratory disease, and ischemic heart disease. There was a consistently high degree of heterogeneity among results from primary studies, which was only partially accounted for in meta-regression. The magnitude of the observed association was larger in studies with probably high or high risk of bias in the confounding domain, and pooled estimates were substantially smaller for cardiovascular, respiratory and cerebrovascular mortality when these studies were excluded. Heterogeneity based on $I^2$ was reduced substantially for cerebrovascular and respiratory mortality after excluding these studies, which we believe is related to the combination of a smaller number of studies and exclusion of studies reporting associations of a magnitude that differed substantially from the remaining studies. We also found that pooled estimates were sensitive to which covariates were considered critical potential confounders; pooled estimates for cardiovascular and cerebrovascular mortality were considerably larger when BMI rather than smoking was treated as a critical potential confounder, as specified in the WHO risk of bias criteria. We did not find an association between log HR and risk of bias in the exposure assessment domain, although one would expect that greater exposure measurement error (likely non-differential) would tend to bias associations towards the null, depending on the type of error [70–73]. Indeed, Crouse et al. reported that the magnitude of the association was larger when change of residential address was accounted for [59]. There was evidence of publication bias based on tests of funnel plot asymmetry for all/ natural cause and lung cancer mortality, but application of trim and fill revealed only small reductions in pooled estimates of association. The pooled estimate for all/natural cause mortality based on single pollutant models was larger than that based on multi-pollutant models, and confidence intervals on the multi-pollutant pooled estimate overlapped 1 or no association. Of the approximately one-third of studies that examined the shape of the exposure response relationship, most concluded that it was linear.

There have been five previous systematic reviews/meta-analyses of the association between long term $NO_2$ exposure and mortality in cohort studies. Pooled estimates from these studies are summarized in Table 3 in comparison with the present study, by cause of death. With the exception of the recent paper by Huangfu and Atkinson [20], for all four categories of cause of death, pooled estimates from the present study were based on approximately twice as many primary studies as the most recent previous systematic review. Pooled estimates varied most for cardiovascular and respiratory mortality, while they were relatively consistent in magnitude for all/natural cause and lung cancer mortality. Our pooled estimate was the largest for respiratory mortality based on all studies, but consistent with lower pooled estimates from other studies after excluding those with probably high or high risk of bias in the confounding domain. Heterogeneity among results from primary studies was uniformly high in earlier meta-analyses, with the exception of one pooled estimate for respiratory mortality based on only 8 primary studies. Only Huangfu and Atkinson [20], conducted an assessment of risk of bias over multiple domains, but neither they nor the other systematic reviews provided pooled estimates from paired single and multi-pollutant models from the same primary studies. In subgroup analyses, Atkinson et al. found that pooled estimates of associations with all/ natural cause, cardiovascular and lung cancer mortality based on cohorts restricted to older age groups were larger than those based on all ages [19]. Additionally, those based on models adjusting

**Table 3. Comparison of pooled estimates from current study and previous meta-analyses.**

| Cause of death | Author | Central[b] | Lower 95%CI | Upper 95%CI | n | I² (%) |
|---|---|---|---|---|---|---|
| **All/Natural cause** | Stieb 2020 | 1.064 | 1.034 | 1.094 | 39 | 99 |
| | Stieb 2020[a] | 1.047 | 1.023 | 1.072 | 32 | 97 |
| | Huangfu 2020 | 1.038 | 1.010 | 1.067 | 24 | 97 |
| | Atkinson 2018 | 1.038 | 1.019 | 1.057 | 20 | 84 |
| | Faustini 2014 | 1.079 | 1.036 | 1.124 | 12 | 89 |
| | Hoek 2013 | 1.106 | 1.059 | 1.156 | 12 | 73 |
| **Cardiovascular** | Stieb 2020 | 1.139 | 0.997 | 1.301 | 29 | 100 |
| | Stieb 2020[a] | 1.058 | 1.026 | 1.091 | 23 | 93 |
| | Faustini 2014 | 1.265 | 1.172 | 1.365 | 16 | 98 |
| | Atkinson 2018 | 1.057 | 1.038 | 1.096 | 15 | 83 |
| **Respiratory** | Stieb 2020 | 1.174 | 1.016 | 1.357 | 29 | 100 |
| | Stieb 2020[a] | 1.062 | 1.035 | 1.089 | 24 | 66 |
| | Huangfu 2020 | 1.057 | 1.019 | 1.097 | 15 | 83 |
| | Atkinson 2018 | 1.057 | 1.019 | 1.096 | 13 | 76 |
| | Faustini 2014 | 1.046 | 1.032 | 1.061 | 8 | 0 |
| **Lung cancer** | Stieb 2020 | 1.084 | 1.045 | 1.124 | 28 | 97 |
| | Stieb 2020[a] | 1.083 | 1.041 | 1.126 | 23 | 85 |
| | Atkinson 2018 | 1.096 | 1.038 | 1.156 | 16 | 88 |
| | Hamra 2015 | 1.077 | 1.019 | 1.156 | 15 | 73 |

[a]After exclusion of studies with probably high or high risk of bias in confounding domain.

[b]Per 10 ppb.

for key confounders (BMI, smoking) were smaller than those without these adjustments (natural cause, cardiovascular, respiratory and lung cancer mortality), similar to our findings; and those for natural cause and cardiovascular mortality based on residential exposure estimates were larger than those based on area level exposures [19]. In Faustini et al's systematic review, the effects of excluding studies based on individuals with pre-existing disease, older age groups, or with area level exposure characterization were inconsistent [17]. Hamra et al. found little impact of method of exposure characterization or presence or absence of adjustment for selected confounders (with the exception of studies including adjustment for occupation, which generated a smaller pooled estimate) [18]. Huangfu and Atkinson also found little impact of adjustment for selected confounders [20]. They concluded that the certainty of evidence was moderate for all-cause, respiratory and acute lower respiratory infection mortality, and high for chronic obstructive pulmonary disease mortality [20].

## Other lines of evidence

The 2016 US Environmental Protection Agency (EPA) Integrated Science Assessment on oxides of nitrogen concluded that toxicological evidence suggested there were several possible mechanisms through which long term exposure to $NO_2$ could contribute to adverse respiratory and cardiovascular effects, including pulmonary inflammation, oxidative stress, pulmonary injury, changes in lung morphology, impaired respiratory immune defences, atherosclerosis, autonomic dysfunction, and changes in lipid metabolism [11]. Examples of findings from relevant studies include the modulation of alveolar macrophage activity in rats in association with $NO_2$ exposure [74], and increased mortality from respiratory infection in mice in two older studies of $NO_2$ exposure [75, 76]. With respect to cardiovascular effects, in

one study, long term exposure to $NO_2$ was associated with reduced heart rate variability, but only in women [77]. In a Spanish cross sectional study, long term $NO_2$ exposure was associated with increased blood pressure [78] and subclinical atherosclerosis [79]. A more recent study in Germany found that those with higher long term $NO_2$ exposure had significantly higher odds of elevated ankle-brachial index, reflective of arterial stiffening, although the magnitude of the association was reduced in a joint model with $PM_{2.5}$ absorbance (a proxy for elemental carbon) [80]. Overall, however, the evidence base was considered limited, associations were inconsistent, and it was difficult to separate effects of $NO_2$ from those of co-pollutants [11]. With respect to short term exposure to $NO_2$ and mortality, a recent systematic review concluded that there was a high degree of certainty of the evidence linking 24 hour average exposure and mortality [81]. While there may be a degree of overlap in the effects captured by studies of short term and long term exposure, the overall health impact captured by each type of design is not identical [82], and certainty of evidence regarding effects of short term exposure does not necessarily imply certainty regarding effects of long term exposure.

## Overall rating of quality and strength of evidence

In their 2016 Science Assessments, both the US EPA and Health Canada concluded that the evidence was suggestive of, but not sufficient to infer, a causal relationship between long term $NO_2$ exposure and mortality, based on a smaller number of studies, and fewer studies examining the impact of adjustment for co-pollutants than considered here, as well as limited and inconsistent supporting mechanistic evidence from human and animal studies [11, 12]. Applying the Navigation Guide methodology [33] and the causality determination framework used by the US EPA and Health Canada [12] to our current findings, several factors are considered downgrading factors in interpreting the overall quality of evidence. These include the significant heterogeneity among studies even after accounting for sources of heterogeneity, and the relatively large proportion of studies rated as probably high or high risk of exposure assessment bias (57.1%) (even though presence of this bias was not associated with magnitude of association in meta-regression). There was also evidence of publication bias, particularly for lung cancer mortality, but it did not have a substantial impact on pooled estimates of association. Risk of bias from residual confounding was evaluated both in relation to inclusion of critical potential confounders in statistical models, as well as impacts of co-pollutants and other co-exposures on the magnitude of associations. Pooled estimates indicated that $NO_2$ remained significantly associated with all/natural cause, cardiovascular, lung cancer, respiratory and ischemic heart disease mortality after exclusion of 12 studies with probably high or high risk of bias in the confounding domain. However, after excluding these studies, only 9 studies of all/natural cause mortality provided estimates based on both single and multi-pollutant models, and the pooled estimate indicated that $NO_2$ was no longer significantly associated with mortality after adjusting for co-pollutants. Fewer paired estimates from single and multi-pollutant models were available for other causes of death. Pooled estimates from multi-pollutant models were consistently smaller in magnitude than those from single pollutant models, and the pooled estimates based on multi-pollutant models included 1, indicating no association, with the exception of ischemic heart disease mortality, which was based on only three studies. Multi-pollutant models should be interpreted with caution in that the sensitivity of the effect of one pollutant to inclusion of other pollutants in a joint model is affected by factors such as the correlation among pollutants and their relative degree of exposure measurement error [83]. There is nonetheless evidence of confounding by co-pollutants of the association of long term $NO_2$ exposure with mortality. Few studies jointly modelled $NO_2$ with traffic noise. In a recent review, Tétrault et al. concluded that cardiovascular effects of long term air pollution

exposure were probably independent of noise, but this was based on only nine studies, including only one study of air pollution and mortality [15]. One study found that that the association of NO$_2$ with mortality was reduced in magnitude in models with both noise and/or green space [54]. Specifically with respect to cerebrovascular mortality, imprecision is also considered a downgrading factor, as the 95% CI on the pooled estimate overlapped the null. In contrast to these downgrading factors, characterization of the exposure-response relationship as linear, supralinear, or linear with a threshold in 27 of the 29 studies in which this was evaluated, is considered an upgrading factor applicable to all six causes of death. Huangfu and Atkinson also downgraded the evidence in relation to heterogeneity for all cause and respiratory mortality, and upgraded it in relation to evidence of an exposure-response relationship [20]. Based on these considerations, we conclude that for all causes of death other than cerebrovascular disease, the overall quality of the evidence from cohort studies is moderate, the strength of evidence of toxicity is limited, and the overall evidence continues to be suggestive of, but not sufficient to infer, a causal relationship between long term NO$_2$ exposure and mortality. In the case of cerebrovascular disease mortality, owing to the smaller number of primary studies and the non-significant smaller magnitude association based on the pooled estimate, we conclude that the overall quality of the evidence from cohort studies is low, the strength of evidence is inadequate, and the overall evidence is inadequate to infer a causal relationship. Upgrading to a conclusion that there is sufficient evidence for a causal relationship would require more conclusive evidence ruling out potential confounders as well as consistent supporting animal toxicological and human clinical evidence. Future studies could address uncertainties related to confounding by co-pollutants by more consistently examining their correlations and effects in multi-pollutant models, in particular adjusting for other traffic-related pollutants and concomitant exposures like noise, green space and stress. Only about one third of the studies we reviewed addressed the shape of the concentration-response relationship, therefore examination of this issue in future studies would also be informative. While the evidence reviewed does not support the unequivocal conclusion that long term exposure to outdoor NO$_2$ causes an increased risk of death, identifying the true causal agent is of major importance to public health. Vehicle emissions are one of the main sources of NO$_2$, but vehicle emissions, and secondary pollutants arising from vehicle emissions also include numerous other potentially toxic pollutants such as carbon monoxide, particulate matter, benzene, formaldehyde, acetaldehyde, 1,3-butadiene, ozone, nitrates and organic and inorganic acids [10]. If the true causal agent is not NO$_2$, control measures which specifically reduce NO$_2$ will not reduce mortality risks. Conversely, identification and control of the true causal agent will have considerable public health benefits.

## Conclusions

We conducted a synthesis of the evidence from 79 cohort studies examining the association between long term NO$_2$ exposure and natural cause and cause-specific mortality, including sensitivity analyses based on pooling method, leave one out analysis and trim and fill, meta-regression to examine sources of heterogeneity, and analysis of single vs. multi-pollutant models. We concluded that for all causes of death other than cerebrovascular disease, the overall quality of the evidence is moderate and the strength of evidence of toxicity was categorized as limited, while for cerebrovascular disease the overall quality of the evidence is low, and strength of evidence was rated inadequate. Important uncertainties remain, including potential confounding by co-pollutants or other concomitant exposures, and limited supporting mechanistic evidence. Identification and control of the true causal agent linking long term NO$_2$ exposure and mortality, whether NO$_2$ itself or another correlated exposure, will have considerable public health benefits.

## Supporting information

**S1 Fig. Risk of bias criteria.**
(PDF)

**S2 Fig. Risk of bias ratings for individual studies.**
(PDF)

**S3 Fig. Forest plots by region.**
(PDF)

**S4 Fig. Hazard ratios from single pollutant models from individual cohort studies and pooled estimates by region (specific causes of death).**
(PDF)

**S5 Fig. Funnel plot of log(hazard ratio) vs. standard error, lung cancer mortality.**
(PDF)

**S1 File. Review protocol.**
(PDF)

**S2 File. Data.**
(CSV)

**S3 File. PRISMA 2009 checklist.**
(PDF)

**S1 Table. Details of search strategies.**
(PDF)

**S2 Table. Navigation guide criteria for overall quality and strength of evidence.**
(PDF)

**S3 Table. USEPA/Health Canada criteria for evaluating likelihood of causal relationship.**
(PDF)

**S4 Table. Characteristics of primary studies.**
(PDF)

**S5 Table. Reasons for assigned ratings of risk of bias greater than low risk (or unable to assess).**
(PDF)

**S6 Table. Leave one out analyses.**
(PDF)

**S1 Text. Sensitivity to estimator.**
(PDF)

## Acknowledgments

We thank Lisa Glandon, MIS (Health Library, Health Canada) for peer review of the MEDLINE search strategy as well as Tania Onica and Barry Jessiman (Health Canada) for helpful comments. We also thank Drs. Wenqi Gan, Hsien-Ho Lin and Fred Lipfert for their responses to requests for additional information on their studies.

## Author Contributions

**Conceptualization:** David M. Stieb, Carine Zheng, Dina Salama, Robyn Hocking, Ninon Lyrette, Carlyn Matz, Eric Lavigne, Hwashin H. Shin.

**Data curation:** David M. Stieb, Rania Berjawi, Monica Emode, Carine Zheng, Dina Salama, Robyn Hocking, Carlyn Matz.

**Formal analysis:** David M. Stieb, Rania Berjawi, Monica Emode.

**Funding acquisition:** David M. Stieb.

**Methodology:** David M. Stieb, Carine Zheng, Dina Salama, Robyn Hocking, Ninon Lyrette, Carlyn Matz, Eric Lavigne, Hwashin H. Shin.

**Project administration:** David M. Stieb, Ninon Lyrette.

**Supervision:** David M. Stieb.

**Validation:** David M. Stieb, Rania Berjawi, Monica Emode.

**Visualization:** David M. Stieb.

**Writing – original draft:** David M. Stieb.

**Writing – review & editing:** David M. Stieb, Rania Berjawi, Monica Emode, Carine Zheng, Dina Salama, Robyn Hocking, Ninon Lyrette, Carlyn Matz, Eric Lavigne, Hwashin H. Shin.

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
