## [Decision Letter · Decision Letter 0]

4 Nov 2020

PONE-D-20-27306

Systematic review and meta-analysis of cohort studies of long term outdoor nitrogen dioxide exposure and mortality

PLOS ONE

Dear Dr. Stieb,

Thank you for submitting your manuscript to PLOS ONE. After careful consideration, we feel that it has merit but does not fully meet PLOS ONE’s publication criteria as it currently stands. Therefore, we invite you to submit a revised version of the manuscript that addresses all the points raised during the review process.

We look forward to receiving your revised manuscript.

Kind regards,

Gianluigi Forloni

Academic Editor

PLOS ONE

Journal Requirements:

" The funders had no role in study design, data collection and analysis, decision to publish, or preparation of the manuscript."

Reviewers' comments:

Reviewer's Responses to Questions

**Comments to the Author**

1. Is the manuscript technically sound, and do the data support the conclusions?

Reviewer #1: Yes

2. Has the statistical analysis been performed appropriately and rigorously? 

Reviewer #1: Yes

3. Have the authors made all data underlying the findings in their manuscript fully available?

Reviewer #1: Yes

4. Is the manuscript presented in an intelligible fashion and written in standard English?

Reviewer #1: Yes

5. Review Comments to the Author

Reviewer #1: The present manuscript reports the results of a systematic review, aimed to analyse the effects of long-term exposures to ambient NO2 on all-cause and cause-specific mortality, including cardiovascular, respiratory, and cerebrovascular mortality. The study was well conducted, with a rigorous methodology, and a thorough data analysis. The results are interesting, and the interpretation of findings seems to be adequate to extract several highly relevant conclusions for the study field. I have a few concerns I think should be addressed before publication, in order to make some improvements to this already excellent piece of work.

Major comments

My main concern is related to the “rating of quality and strength of evidence” analysis. Two very distinct tools were employed, the Navigation Guide methodology, and the causality determination framework. Within the text, this evaluation was very briefly explained, in fact it was barely mentioned, in the methodology section. There are no results associated with this analysis, and from the mention in the methodology, it jumps directly to the final part of the discussion and conclusions. It is worth noting that this analysis appeared to be relevant enough to be included in the conclusions, and within the abstract.

Given the importance of the strength of evidence evaluation, I guess it deserves further analysis and a better description of the procedures. The main observation is that the authors seem to imply that these two tools were somehow related, but actually they are quite different in the purpose and application. The Navigation Guide is a method for research synthesis in the context of systematic reviews. On the contrary, the causality determination framework is a general framework to consider causality, similar to the Bradford Hill criteria, which includes not only systematic reviews of epidemiological studies but also an overview of different type of studies, based on diverse scientific disciplines.

On one hand, the causality determination framework analysis cannot be performed using the results of this study, it needs many other sources. In this sense, it could be useful to enhance the discussion section, but not for the conclusion section or the abstract, where the main conclusions of this particular study are the ones that should be specifically mentioned.

On the other hand, the methodology lacks a thorough explanation about the way the Navigation Guide was applied. Many if not all the criteria to analyse the strength of evidence need concrete rules to judge the downgrading or upgrading of the level of evidence. This rules were not reported in the text. For example, how large should the magnitude of the effect be, or how “substantial” should the risk of bias be, in order to trigger the downgrades? These rules cannot be understood by reading the Table S2, and they should be clarified in the text or in the table. In addition, the results of the assessment for each criterion/mortality cause should be reported. I have some further observations for specific criteria (see below).

The authors judged the risk of bias criterion (a relatively large proportion of studies rated as probably high or high risk) as sufficient to downgrade the level of evidence. However, it can be seen in Table 1 that the sensitivity analysis excluding articles showing high risk of bias still demonstrate positive and significant HRs for almost all the mortality causes. In this sense, the merely presence of articles showing high risk of bias does not necessarily imply that the evidence is weak, provided that significant pooled HRs can be obtained through a considerable number of articles showing low or probably low risk of bias.

In the same line of thoughts, the high heterogeneity reported for almost all mortality causes could be related to natural variation or true heterogeneity (there are many discussions regarding the real value of the I2 parameter to analyse heterogeneity). If this is true, the observed heterogeneity might have an influence on the estimation (and precision) of the true HR value, but not necessarily on the causal relationship.

In page 10, the authors stated that they excluded studies encompassing the same geographic area or time period. The exact rule for article selection should be reported, e.g. whether broader geographic area or more extended time period was prioritized.

Observing Figure S3 (forest plots), it seems that HR estimates from single-pollutant and multipollutant models from the same articles were included at the same time in the pooled HRs. I’m not sure about it, as this is not clear for me in the text. If this is the case, a problem with double-counting of individual estimates might arise in the pooled estimates.

Another aspect to revise and justify, provided that I am not misinterpreting the procedures, is the combination of different co-pollutants species and different number of co-pollutants in the same pooled estimate.

Minor comments

It is rather surprising the very low value of the I2 for cerebrovascular and respiratory mortality in the sensitivity analysis, as compared to the I2 for the other analyses. At first sight, the difference seems not to be attributable to the number of studies. I think this warrants a mention in the discussion section.

When analysing the exposure-response relationship, the authors found three articles which found evidence of a threshold. These thresholds should be mentioned, as this values are relevant for further analyses, unless they have decided that the results of these studies should not be considered due to some reason. Anyway, I think this should be discussed.

Page 19: “If the true causal agent is not NO2, control measures which specifically reduce NO2 will not reduce mortality risks”.

Comment: this sentence needs further development, i.e. more details should be mentioned regarding which might be specific measures addressed to exclusively reduce NO2. Otherwise, the sentence appears as out-of-place here.

The World Health Organization is carrying out a process to update the global Air Quality Guidelines. To that end, a number of systematic reviews were commissioned to different research groups, with the aim of being an input for the new update of the guidelines. The objective of one of these systematic reviews partially overlaps with the objective of this study, i.e. long-term exposure to nitrogen dioxide and ozone and all-cause and respiratory mortality (see https://www.crd.york.ac.uk/prospero/display_record.php?RecordID=89853 ). That review is going to be included in a special issue ( https://www.sciencedirect.com/journal/environment-international/special-issue/10MTC4W8FXJ ), but as far as I know it is not currently available. Given the relevance and similarities of both reviews, and the work the authors performed in the discussion section comparing this study with previous reviews, perhaps the authors could verify the link, to see if the aforementioned paper become available before the submission of the new version of this manuscript.

6. PLOS authors have the option to publish the peer review history of their article (what does this mean?). If published, this will include your full peer review and any attached files.

Reviewer #1: **Yes: **Pablo Orellano

---

## [Author Response · Author response to Decision Letter 0]

18 Dec 2020

Reviewers' comments:

Reviewer's Responses to Questions

Comments to the Author

1. Is the manuscript technically sound, and do the data support the conclusions?

Reviewer #1: Yes

2. Has the statistical analysis been performed appropriately and rigorously? 

Reviewer #1: Yes

3. Have the authors made all data underlying the findings in their manuscript fully available?

Reviewer #1: Yes

4. Is the manuscript presented in an intelligible fashion and written in standard English?

Reviewer #1: Yes

5. Review Comments to the Author

Reviewer #1: The present manuscript reports the results of a systematic review, aimed to analyse the effects of long-term exposures to ambient NO2 on all-cause and cause-specific mortality, including cardiovascular, respiratory, and cerebrovascular mortality. The study was well conducted, with a rigorous methodology, and a thorough data analysis. The results are interesting, and the interpretation of findings seems to be adequate to extract several highly relevant conclusions for the study field. I have a few concerns I think should be addressed before publication, in order to make some improvements to this already excellent piece of work.

Major comments

My main concern is related to the “rating of quality and strength of evidence” analysis. Two very distinct tools were employed, the Navigation Guide methodology, and the causality determination framework. Within the text, this evaluation was very briefly explained, in fact it was barely mentioned, in the methodology section. There are no results associated with this analysis, and from the mention in the methodology, it jumps directly to the final part of the discussion and conclusions. It is worth noting that this analysis appeared to be relevant enough to be included in the conclusions, and within the abstract.

Given the importance of the strength of evidence evaluation, I guess it deserves further analysis and a better description of the procedures. The main observation is that the authors seem to imply that these two tools were somehow related, but actually they are quite different in the purpose and application. The Navigation Guide is a method for research synthesis in the context of systematic reviews. On the contrary, the causality determination framework is a general framework to consider causality, similar to the Bradford Hill criteria, which includes not only systematic reviews of epidemiological studies but also an overview of different type of studies, based on diverse scientific disciplines.

On one hand, the causality determination framework analysis cannot be performed using the results of this study, it needs many other sources. In this sense, it could be useful to enhance the discussion section, but not for the conclusion section or the abstract, where the main conclusions of this particular study are the ones that should be specifically mentioned.

On the other hand, the methodology lacks a thorough explanation about the way the Navigation Guide was applied. Many if not all the criteria to analyse the strength of evidence need concrete rules to judge the downgrading or upgrading of the level of evidence. This rules were not reported in the text. For example, how large should the magnitude of the effect be, or how “substantial” should the risk of bias be, in order to trigger the downgrades? These rules cannot be understood by reading the Table S2, and they should be clarified in the text or in the table. In addition, the results of the assessment for each criterion/mortality cause should be reported. I have some further observations for specific criteria (see below).

RESPONSE: We thank Dr. Orellano for these detailed and thoughtful comments. We agree that this aspect of our paper was not sufficiently transparent and have added further details to the methods and discussion sections and revised the abstract. Note that we have opted to report our conclusions on the overall quality and strength of the evidence, as well as likelihood of causal relationships, in the discussion section, as we consider this interpretation of our findings.

First, we have expanded the methods section under “Overall rating of quality and strength of evidence,” as follows. “We applied the Navigation Guide methodology [33] and the causality determination framework used by the US EPA and Health Canada [12] (S2, S3 Tables) to judge the overall quality and strength of the evidence, and likelihood of a causal relationship. Following the Navigation Guide methodology, given the observational nature of the evidence, the starting point for rating overall quality was “moderate,” which was downgraded or upgraded to “low” or “high” based on the criteria summarized in Table S2 [33]. Identification of multiple upgrading or downgrading factors does not necessarily mean that overall quality is upgraded or downgraded multiple levels. Rather, the overall degree of upgrading or downgrading is based on reviewer judgement of the overall importance and impact of all factors considered [33]. The Navigation Guide characterizes strength of evidence of toxicity as “sufficient”, “limited”, “inadequate” or “indicative of a lack of toxicity” (S2 Table), based on the overall quality of the evidence, the direction of effect, confidence in the effect and any other factors identified as germane by the reviewers [33]. Given the parallels with the USEPA/Health Canada causality determination criteria (S3 Table), while we did not conduct a systematic review of other lines of evidence, we drew upon summaries of other lines of evidence from a recent assessment document [11], supplemented by findings from more recent mechanistic studies, in order to characterize the likelihood of a causal relationship.” Regarding concrete rules or criteria for upgrading/downgrading the quality of evidence, additional details are available from reference 33. However, the Navigation Guide does not prescribe quantitative criteria. 

Second, we have also added to the discussion section as follows. “Specifically with respect to cerebrovascular mortality, imprecision is also considered a downgrading factor, as the 95% CI on the pooled estimate overlapped the null….we conclude that for all causes of death other than cerebrovascular disease, the overall quality of the evidence from cohort studies is moderate, the strength of evidence of toxicity is limited, and the overall evidence continues to be suggestive of, but not sufficient to infer, a causal relationship between long term NO2 exposure and mortality. In the case of cerebrovascular disease mortality, owing to the smaller number of primary studies and the non-significant smaller magnitude association based on the pooled estimate, we conclude that the overall quality of the evidence from cohort studies is low, the strength of evidence is inadequate, and the overall evidence is inadequate to infer a causal relationship.” 

Finally, we have changed the concluding statements in the abstract as follows. “For all causes of death other than cerebrovascular disease, the overall quality of the evidence is moderate, and the strength of evidence is limited, while for cerebrovascular disease, overall quality is low and strength of evidence is inadequate. Important uncertainties remain, including potential confounding by co-pollutants or other concomitant exposures, and limited supporting mechanistic evidence.”

The authors judged the risk of bias criterion (a relatively large proportion of studies rated as probably high or high risk) as sufficient to downgrade the level of evidence. However, it can be seen in Table 1 that the sensitivity analysis excluding articles showing high risk of bias still demonstrate positive and significant HRs for almost all the mortality causes. In this sense, the merely presence of articles showing high risk of bias does not necessarily imply that the evidence is weak, provided that significant pooled HRs can be obtained through a considerable number of articles showing low or probably low risk of bias.

RESPONSE: We have clarified our reasoning on the issue of confounding in this section of the discussion, noting that, “Risk of bias from residual confounding was evaluated both in relation to inclusion of critical potential confounders in statistical models, as well as impacts of co-pollutants and other co-exposures on the magnitude of associations. Pooled estimates indicated that NO2 remained significantly associated with all/natural cause, cardiovascular, lung cancer, respiratory and ischemic heart disease mortality after exclusion of 12 studies with probably high or high risk of bias in the confounding domain. However, after excluding these studies, only 9 studies of all/natural cause mortality provided estimates based on both single and multi-pollutant models, and the pooled estimate indicated that NO2 was no longer significantly associated with mortality after adjusting for co-pollutants. Fewer paired estimates from single and multi-pollutant models were available for other causes of death. Pooled estimates from multi-pollutant models were consistently smaller in magnitude than those from single pollutant models, and the pooled estimates based on multi-pollutant models included 1, indicating no association, with the exception of ischemic heart disease mortality which was based on only three studies. Multi-pollutant models should be interpreted with caution in that the sensitivity of the effect of one pollutant to inclusion of other pollutants in a joint model is affected by factors such as the correlation among pollutants and their relative degree of exposure measurement error [81]. There is nonetheless evidence of confounding by co-pollutants of the association of long term NO2 exposure with mortality.” We have also noted that it is possible that associations of NO2 with mortality are confounded by noise and green space, but few studies have evaluated this. 

In the same line of thoughts, the high heterogeneity reported for almost all mortality causes could be related to natural variation or true heterogeneity (there are many discussions regarding the real value of the I2 parameter to analyse heterogeneity). If this is true, the observed heterogeneity might have an influence on the estimation (and precision) of the true HR value, but not necessarily on the causal relationship.

RESPONSE: Dr. Orellano is correct that high heterogeneity could simply reflect natural variation among studies. Nonetheless, inconsistency/heterogeneity is clearly identified as a downgrading factor in the Navigation Guide, and as we emphasize in this section of the discussion, it could not be explained by the various factors we considered in meta-regression. We note that the new systematic review by Huangfu and Atkinson (2020) that Dr. Orellano mentioned in his final comment, also downgraded the evidence for both all cause and respiratory mortality in relation to inconsistency/heterogeneity. We have added a statement about Huangfu and Atkinson’s conclusion on this issue to the discussion.

In page 10, the authors stated that they excluded studies encompassing the same geographic area or time period. The exact rule for article selection should be reported, e.g. whether broader geographic area or more extended time period was prioritized.

RESPONSE: We have added shading to supplementary table S4 to indicate which papers were selected (n=53) vs. not selected in relation to overlapping time periods and geographic areas (n=21), as well as results only being available from multi-pollutant models (n=1), or too few studies of the outcome (n=4). Note that in reviewing our exclusions we made two changes, opting to include results from an additional two studies (five effect size estimates) of COPD mortality in the pooled estimate for respiratory mortality, and results from an additional two studies of myocardial infarction mortality in the pooled estimate for ischemic heart disease mortality. These changes did not materially affect the pooled estimates. 

Observing Figure S3 (forest plots), it seems that HR estimates from single-pollutant and multipollutant models from the same articles were included at the same time in the pooled HRs. I’m not sure about it, as this is not clear for me in the text. If this is the case, a problem with double-counting of individual estimates might arise in the pooled estimates.

RESPONSE: There are no pooled estimates associated with Figure S3 forest plots, thus there is no potential for double counting. In keeping with PRISMA guidelines, these plots simply show all extracted results for all included papers, including results for population subgroups and results from single and multi-pollutant models. 

Another aspect to revise and justify, provided that I am not misinterpreting the procedures, is the combination of different co-pollutants species and different number of co-pollutants in the same pooled estimate.

RESPONSE: If the reviewer is referring to Figure S3, the same response applies as for the previous comment, i.e. there are no pooled estimates associated with these forest plots. If the reviewer is referring to Figure 3, as explained in the Methods section pertaining to data extraction, “Results from multi-pollutant models that resulted in the greatest reduction in magnitude of effect compared to single pollutant results were selected in order to bracket the magnitude of effect from each study.” We have further clarified in the methods section that, “Our objective was not to assess the magnitude of the association with NO2 in the presence of a single co-pollutant common to all studies, but rather to determine the maximum potential for confounding of the association of mortality with NO2 by any co-pollutant(s).”

Minor comments

It is rather surprising the very low value of the I2 for cerebrovascular and respiratory mortality in the sensitivity analysis, as compared to the I2 for the other analyses. At first sight, the difference seems not to be attributable to the number of studies. I think this warrants a mention in the discussion section.

RESPONSE: We have added a statement in the discussion indicating that we believe this is related to the combination of a smaller number of studies and exclusion of studies reporting associations of a magnitude that differed substantially from the remaining studies.

When analysing the exposure-response relationship, the authors found three articles which found evidence of a threshold. These thresholds should be mentioned, as this values are relevant for further analyses, unless they have decided that the results of these studies should not be considered due to some reason. Anyway, I think this should be discussed.

RESPONSE: We have clarified in this section that of three studies finding evidence of a threshold, only one study identified specific numeric values, ranging from 20-40 µg/m3, depending on age and cause of death.

Page 19: “If the true causal agent is not NO2, control measures which specifically reduce NO2 will not reduce mortality risks”.

Comment: this sentence needs further development, i.e. more details should be mentioned regarding which might be specific measures addressed to exclusively reduce NO2. Otherwise, the sentence appears as out-of-place here.

RESPONSE: We have added further explanatory text indicating that, “Vehicle emissions are one of the main sources of NO2, but vehicle emissions, and secondary pollutants arising from vehicle emissions, also include numerous other potentially toxic pollutants such as carbon monoxide, particulate matter, benzene, formaldehyde, acetaldehyde, 1,3-butadiene, ozone, nitrates and organic and inorganic acids.” Thus, if a pollutant control technology only reduces NO2, but NO2 is not primarily responsible for adverse health effects, anticipated health benefits of pollution control measures will not be realized. 

The World Health Organization is carrying out a process to update the global Air Quality Guidelines. To that end, a number of systematic reviews were commissioned to different research groups, with the aim of being an input for the new update of the guidelines. The objective of one of these systematic reviews partially overlaps with the objective of this study, i.e. long-term exposure to nitrogen dioxide and ozone and all-cause and respiratory mortality (see https://www.crd.york.ac.uk/prospero/display_record.php?RecordID=89853 ). That review is going to be included in a special issue ( https://www.sciencedirect.com/journal/environment-international/special-issue/10MTC4W8FXJ ), but as far as I know it is not currently available. Given the relevance and similarities of both reviews, and the work the authors performed in the discussion section comparing this study with previous reviews, perhaps the authors could verify the link, to see if the aforementioned paper become available before the submission of the new version of this manuscript.

RESPONSE: We thank Dr. Orellano for noting this and we have now included this paper in our summary of other systematic reviews and meta-analyses. We also noted two studies (Hartiala et al. 2016, Desikan et al. 2016) that were included in this review which were not included in our analysis. We have added these, but they did not materially affect our results.

---

## [Decision Letter · Decision Letter 1]

20 Jan 2021

Systematic review and meta-analysis of cohort studies of long term outdoor nitrogen dioxide exposure and mortality

PONE-D-20-27306R1

Dear Dr. Stieb,

We’re pleased to inform you that your manuscript has been judged scientifically suitable for publication and will be formally accepted for publication once it meets all outstanding technical requirements.

Kind regards,

Gianluigi Forloni

Academic Editor

PLOS ONE

Additional Editor Comments (optional):

Reviewers' comments:

Reviewer's Responses to Questions

**Comments to the Author**

1. If the authors have adequately addressed your comments raised in a previous round of review and you feel that this manuscript is now acceptable for publication, you may indicate that here to bypass the “Comments to the Author” section, enter your conflict of interest statement in the “Confidential to Editor” section, and submit your "Accept" recommendation.

Reviewer #1: All comments have been addressed

2. Is the manuscript technically sound, and do the data support the conclusions?

Reviewer #1: Yes

3. Has the statistical analysis been performed appropriately and rigorously? 

Reviewer #1: Yes

4. Have the authors made all data underlying the findings in their manuscript fully available?

Reviewer #1: Yes

5. Is the manuscript presented in an intelligible fashion and written in standard English?

Reviewer #1: Yes

6. Review Comments to the Author

Reviewer #1: In this submission, all my concerns were carefully addressed. The issues that seemed unclear in the previous round of revision were sufficiently explained, and the suggested modifications were incorporated. Thus, in my opinion this new version of the manuscript is suitable for publication in PLOS ONE. I congratulate Dr. Stieb and colleagues for the outstanding work performed for this systematic review.

7. PLOS authors have the option to publish the peer review history of their article (what does this mean?). If published, this will include your full peer review and any attached files.

Reviewer #1: **Yes: **Pablo Orellano

---

## [Editor Report · Acceptance letter]

25 Jan 2021

PONE-D-20-27306R1 

Systematic review and meta-analysis of cohort studies of long term outdoor nitrogen dioxide exposure and mortality 

Dear Dr. Stieb:

I'm pleased to inform you that your manuscript has been deemed suitable for publication in PLOS ONE. Congratulations! Your manuscript is now with our production department. 

Kind regards, 

on behalf of

Dr. Gianluigi Forloni 

Academic Editor

PLOS ONE